# Sample size calculation for phylogenetic case linkage

**Shirlee Wohl, John R. Giles⬚, Justin Lessler⬚** *

Johns Hopkins Bloomberg School of Public Health, Department of Epidemiology, Baltimore, Maryland, United States of America

* justin@jhu.edu

## Abstract

Sample size calculations are an essential component of the design and evaluation of scientific studies. However, there is a lack of clear guidance for determining the sample size needed for phylogenetic studies, which are becoming an essential part of studying pathogen transmission. We introduce a statistical framework for determining the number of true infector-infectee transmission pairs identified by a phylogenetic study, given the size and population coverage of that study. We then show how characteristics of the criteria used to determine linkage and aspects of the study design can influence our ability to correctly identify transmission links, in sometimes counterintuitive ways. We test the overall approach using outbreak simulations and provide guidance for calculating the sensitivity and specificity of the linkage criteria, the key inputs to our approach. The framework is freely available as the R package *phylosamp*, and is broadly applicable to designing and evaluating a wide array of pathogen phylogenetic studies.

**Data Availability Statement:** All code and simulation data are available at: https://github.com/HopkinsIDD/phylosamplesize.

**Funding:** Funding was provided by Bill and Melinda Gates Foundation OPP1195157 (S.W. and J.L.). The funders had no role in study design, data

## Author summary

Sequencing the genetic material of viral and bacterial pathogens has become an important part of tracking and combating human infectious diseases. Specifically, comparing the pathogen DNA or RNA sequences collected from infected individuals can allow researchers and public health experts to determine who infected whom, or detect when a pathogen entered a specific country or geographic area. However, it is often impossible to collect samples from every single infected person, and these missing sequences can pose problems for this type of analysis, especially if there is some bias behind which samples were selected for sequencing. We have developed a mathematical framework that allows users to determine the probability their conclusions about pathogen transmission are correct given the number and proportion of samples from a pathogen outbreak they have sequenced. This framework is freely available, easy to use, and broadly generalizable to any pathogen, and we hope that it can be used to inform the design and sampling strategies behind future sequencing-based studies.

collection and analysis, decision to publish, or preparation of the manuscript.

**Competing interests:** The authors have declared that no competing interests exist.

## Introduction

As the cost of pathogen sequencing has declined, the number and size of studies based on pathogen sequence analysis has increased dramatically [1]. Traditionally, researchers have sequenced convenience samples collected as part of routine clinical or public health activities (e.g., diagnostic specimens collected as part of an outbreak response), or as part of studies where specimens are collected for other purposes. However, the analysis of pathogen genomic sequences is increasingly becoming a primary goal of both research studies and public health surveillance efforts [2–5]. This shift has been driven by the apparent utility of pathogen sequence data for understanding aspects of pathogen spread ranging from the frequency and source of introductions into a region [6–10], to identifying endogenous spread of emerging diseases [11,12], to understanding the role of "hotspots" in maintaining broader community epidemics [13], to understanding transmission patterns at an individual or "microscale" level [3,14].

Despite these many examples, there is a lack of clear and accessible guidance for appropriately designing and sizing studies aimed at understanding pathogen transmission, or for evaluating the design and conclusions of past studies. Without such guidance, it is difficult for researchers to design studies in a way that maximizes the chances of success, and difficult for reviewers to appropriately evaluate papers and grant applications centered around molecular or phylogenetic outcomes [15,16]. In particular, undersampling or biased sampling can lead to poorly supported inferences about patterns of disease spread [17,18]. While there are examples of researchers conducting careful *a priori* analyses of sampling strategies [19–21], these have largely relied on sophisticated techniques that are not broadly generalizable. Hence, there is a need for broadly accepted and accessible guidance for the selection of specimens for sequencing and phylogenetic analyses.

As noted above, pathogen sequences have been used to understand multiple aspects of infectious disease transmission at scales ranging from the global (e.g., movement of pathogens between countries) to the individual (e.g., reconstruction of individual transmission chains). Arguably, all such analyses can be reduced to the basic question of whether pairs of infected units (individuals, locations, etc.) are related or connected within a particular number of generations of transmission. Therefore, developing tools for assessing the number of sequences needed to confidently identify linked individuals (infections separated by no more than a specific number of generations of transmission) is a good place to start building a theory for power calculations for phylogenetic inference that can later be applied to questions at vastly different spatial or temporal scales. In this paper, we present a framework for making critical decisions about study design when the goal is to identify infector-infectee pairs, and we illustrate this approach with simulation studies.

## Methods

### General principles

In this paper we will focus on studies that aim to identify infector-infectee pairs from phylogenetic analysis of pathogen sequence data collected from infected individuals. We assume the study aims to achieve some level of certainty that identified infector-infectee pairs are correct, and may also require identification of some minimum number of pairs. Below we lay out a precise terminology (**Table 1**) and general principles.

To start, we define the term *linkage criteria* to represent all the criteria used to infer whether a set of infected individuals are linked to one another by direct transmission. The *linkage criteria* can be derived from a combination of genetic distance between pathogens isolated from

**Table 1. Parameters used in calculations and simulations.**

| Parameter | Description |
|---|---|
| $M$ | Number of infections sampled |
| $N$ | Total number of (relevant) infected individuals in an outbreak |
| $\rho$ | Proportion of outbreak infections sampled ($M/N$) |
| $\eta$ | Sensitivity of the linkage criteria |
| $\chi$ | Specificity of the linkage criteria |
| $\phi$ | Probability that an identified link represents a true transmission event (1-False Discovery Rate) |
| $R$ | Reproductive number of a pathogen |
| $R_{\text{pop}}$ | Average reproductive number of a pathogen in a finite population (always <1) |
| $\mu$ | Substitution rate of the pathogen (in substitutions observed per genome per transmission event) |

different individuals, tree structure (e.g., clade support), and epidemiologic information (e.g., relative dates of symptom onset). We refer to infections inferred to be connected by transmission using this criteria as *linked pairs*. Some of these linked pairs will represent actual transmission events (*true transmission pairs*) and some will be false positives. We want to determine the sample size ($M$) and proportion of the population ($\rho$) required to recover a predetermined number of linked pairs, while keeping the *false discovery rate* (the proportion of these linked pairs that are false positives) below a predetermined threshold. When applied to a study where design was dictated by other factors (e.g., specimen availability), the same methods can be used to determine the *false discovery rate*, which will inform the confidence we have in any conclusions about disease transmission in that study.

To capture *true transmission pairs*, the infector and their partner infectee must both be in the sample. Therefore, correctly identifying direct transmission links (and, conversely, calculating the false discovery rate) depends on the sampling fraction ($\rho$), which is equal to the sample size ($M$) divided by the total number of infected individuals in the relevant population ($N$). Identification of these links will further depend on the *sensitivity* ($\eta$) and *specificity* ($\chi$) of the criteria used to define linkage. We define sensitivity as the probability that the linkage criteria will identify a true transmission pair as a linked pair given that both the infector and infectee are in the sample. Similarly, the specificity is the probability that two infections not linked by transmission are not linked by the linkage criteria.

Here we show that, if we have reasonable estimates of the sampling fraction, sensitivity, and specificity, we can, for a sample of size $M$, estimate the false discovery rate. The relationship between these parameters can then be used to design studies with a sample size and sampling fraction that minimizes the false discovery rate and therefore maximizes our ability to draw inferences from identified infections.

## Calculating sample size and false discovery rate

**Multiple links and multiple true transmissions.** In most transmission scenarios, we will be interested in linking an infected individual to both their infector and anyone they infect. Therefore, we must account for the fact that each infection in an outbreak may be linked by transmission to multiple other infections, only some of which may have been sampled. If the goal is to identify all true transmission pairs in the sample, the linkage criteria used must similarly allow for each infection to be linked to multiple other infections. Given this, we can calculate the probability of correctly identifying a true transmission pair, $\phi$ (equal to one minus the false discovery rate), as a function of just the sensitivity and specificity of the linkage criteria, the proportion sampled, and the sample size. Conceptually, this probability of correctly

identifying a transmission pair is equal to the number of true positives (correctly identified true transmission pairs) divided by the total number of positives (linked pairs, regardless of true transmission status):

$$\phi = \frac{\text{True Positives}}{\text{True Positives} + \text{False Positives}}$$

Because we allow each infection to have multiple transmission partners, this probability will also depend on the average number of transmission links per infection, which is determined by the epidemiological parameter $R$, the expected number of other individuals each infected individual infects in a fully susceptible population. However, sampling infections over a finite period of time produces a bounded sampling frame, in which the average number of infectees per infector, denoted $R_{\mathrm{pop}}$, may differ from $R$. This is because terminal nodes in the transmission network within this finite sampling frame are presumed to have no known child infections, and therefore an $R$ value of zero. These nodes (which may or may not have child infections outside the sampling frame) contribute an $R$ value of 0, decreasing the average number of infectees per infector. In fact, $R_{\mathrm{pop}}$ must be less than 1, see 'Estimating the average reproductive number' below. Because each infection is linked to, on average, $R_{\mathrm{pop}}$ infectees as well as its infector, each infection has $R_{pop}+1$ true transmission partners. If we assume that the distribution of the number of transmission partners per infection is Poisson distributed, we get the following equation for the true discovery rate, $\phi$ (see **S1 Text** for full derivation):

$$\phi = \frac{\eta \rho (R_{\mathrm{pop}} + 1)}{\eta \rho (R_{\mathrm{pop}} + 1) + (1 - \chi)(M - \rho(R_{\mathrm{pop}} + 1) - 1)} \tag{1}$$

Under the same assumptions, we show that the total number of sampled true pairs, $\mathbb{E}[\text{number of true pairs}]$, can be calculated as:

$$\mathbb{E}[\text{number of true pairs}] = \frac{M\rho(R_{\mathrm{pop}} + 1)\eta}{2}$$

Through algebraic rearrangement of these equations we can determine the expected number of pairs observed in this sample, $\mathbb{E}[\text{number of pairs observed}]$:

$$\mathbb{E}[\text{number of pairs observed}] = \frac{M}{2}\left[\eta\rho(R_{\mathrm{pop}} + 1) + (1 - \chi)(M - \rho(R_{\mathrm{pop}} + 1) - 1)\right]$$

These equations can be used to determine the false discovery rate $(1-\phi)$ and the expected number of linked pairs given a particular criteria, sample size, and sampling proportion. Additionally, we can use these equations to observe how the expected number of links and the true discovery rate vary with the proportion sampled and the sample size (**Fig 1A**). For a given sensitivity and specificity of the linkage criteria, we observe that the false discovery rate *increases* with sample size if the proportion sampled remains constant, suggesting that studies aimed at correctly identifying the highest proportion of transmission links should prioritize sampling proportion over an arbitrary number of samples. Additionally, the relationship between false discovery rate and sampling proportion is dependent on the sample size needed to obtain that sampling proportion such that the impact of sampling proportion increases with sample size. We also observe the effects of changing sensitivity and specificity on the false discovery rate and find that the specificity of the linkage criteria is of key importance when attempting to minimize the false discovery rate of transmission pairs (**Fig 1B**).

**Single link and single true transmission.** We can also derive the relationship between the sample size and false discovery rate for the special case where each infection is the transmission

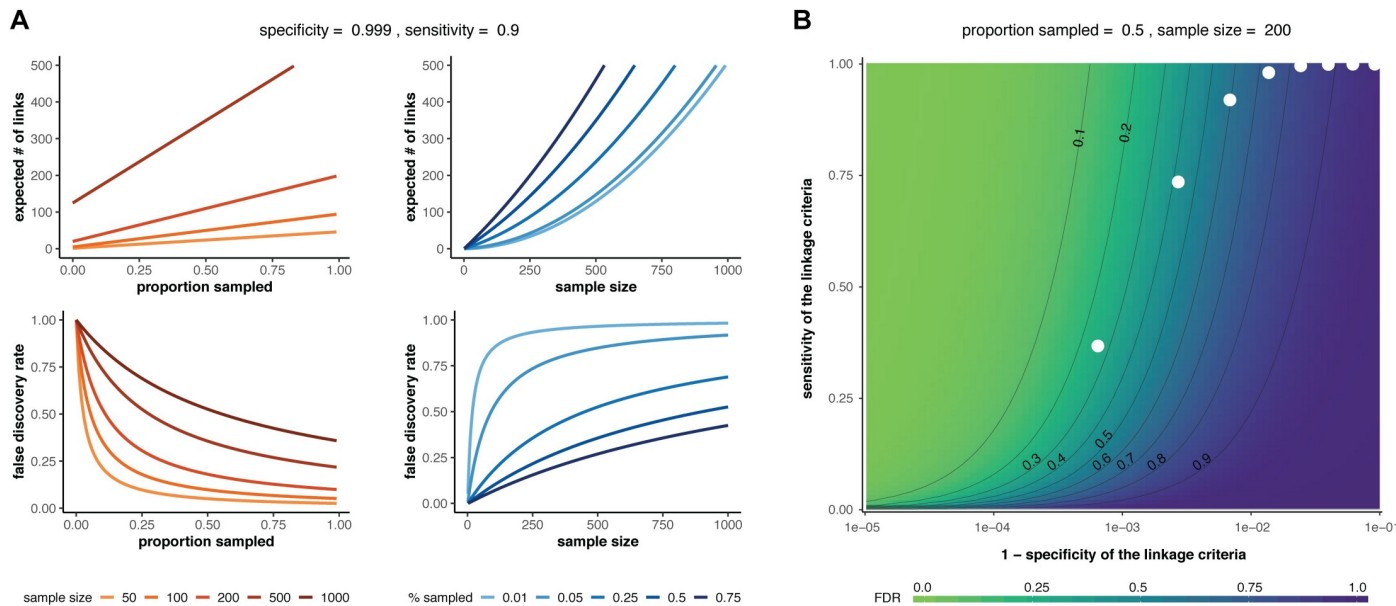

**Fig 1. Sample size and false discovery rate given multiple linkage and multiple transmissions.** (**A**) Effect of sample size (red lines) or proportion sampled (blue lines) on the expected number of linked pairs (upper plots) or the false discovery rate of linked pairs (lower plots). The specificity and sensitivity are held constant. (**B**) Effect of varying the sensitivity and specificity of the linkage criteria on the false discovery rate (FDR). White dots: theoretical sensitivity and specificity values at different genetic distance thresholds (1–10 substitutions between infections; leftmost white dot represents a threshold of 1 substitution) for a hypothetical pathogen with substitution rate = 1 substitution/genome/transmission and $R$ = 2 (see 'Determining sensitivity and specificity' below for details). In both panels, $R_{pop}$ = 1.

pair of exactly one other sample, relevant when we are only interested in identifying the correct infector of a given infection. In this case, the linkage criteria will similarly identify exactly one probable link for each infection [15]. These assumptions about transmission simplify the relationship between sample size and false discovery rate. Here, we calculate the false discovery rate for transmission pairs under these assumptions (see **S1 Text** for full derivation).

The probability of correctly identifying a true transmission pair ($\phi$) under the assumptions of single transmission and single linkage is:

$$\phi = \frac{\eta\rho}{\eta\rho + (1 - \chi^{M-2})(1 - \eta)\rho + (1 - \chi^{M-1})(1 - \rho)} \tag{2}$$

Under the same assumptions, we can also calculate the expected total number of true transmission pairs that will be identified in our sample, $\mathbb{E}[\text{number of true pairs}]$, as:

$$\mathbb{E}[\text{number of true pairs}] = \frac{M}{2}\eta\rho$$

Through algebraic rearrangement of these equations, we can determine the expected number of linked pairs (identified with the linkage criteria) observed in this sample ($\mathbb{E}[\text{number of pairs observed}]$):

$$\mathbb{E}[\text{number of pairs observed}] = \frac{M}{2}[\eta\rho + (1 - \chi^{M-2})(1 - \eta)\rho + (1 - \chi^{M-1})(1 - \rho)]$$

As in the multiple links and multiple transmissions case, we observe that the false discovery rate increases with the sample size, but decreases with the proportion sampled. We also again see the important effect of the specificity of the linkage criteria on the false discovery rate

(**S1 Fig**). The relationships between these parameters and our ability to correctly identify transmission links are clearly robust to transmission model specification.

## Estimating the average reproductive number

In the previous section, we distinguished $R$, the basic reproductive number of a pathogen, from $R_{pop}$, the *average* reproductive number in a bounded sampling frame. This is an important distinction because we can show that the average reproductive number ($R_{pop}$) is at most one. This is because any sampling frame contains a finite number of infected individuals, and individuals on terminal nodes of the captured transmission chain have not, by definition, infected any other individuals within the sampling frame (though they may have passed the infection to others outside the finite sample). Averaging the $R$ value from these terminal nodes (which is zero, because they are terminal nodes) with the R value from all other nodes is what allows the $R_{pop}$ average to drop below one, even when the true value of R is significantly greater than one. In other words, there will always be more infections (at minimum, all infectees in a transmission chain plus a single index case) than infection events (see **S2A Fig**). Hence, $R_{pop}$, which is equal to the number of actual transmission events divided by the number of infections, will be at most one.

In epidemic situations where there is a single introduction, $R_{pop}$ will be close to one, as the number of infections will exceed the number of infection events by precisely one. In situations where there are multiple introductions (e.g., transmission chains that are persistently seeded from sources outside the sampling frame) then $R_{pop}$ may be substantially less than one (**S2B Fig**). Specifically:

$$\frac{\text{cases} - \text{introductions}}{\text{cases}}$$

The examples shown in this paper focus on epidemics seeded by a single introduction, where $R_{pop}$ is approximately equal to one.

## Determining sensitivity and specificity

In the framework presented here, the sensitivity and specificity of the linkage criteria are needed to estimate the false discovery rate from sample size and vice versa. This criteria can be based on a number of phylogenetic and epidemiological metrics, and may depend on the data available for a particular study. In this section, we outline two methods for approximating the sensitivity and specificity of a simple genomic metric: genetic distance.

Both methods involve determining these parameters from the discrete distributions of genetic distances between linked and unlinked infections, but they differ in how these distributions are obtained. Given the distributions, we can consider a number of different genetic distance thresholds (e.g., 1 or 2 mutations observed between sequences) that could be used as the criteria for differentiating between linked and unliked pairs, and we can calculate the sensitivity and specificity at each. The optimal threshold and its associated sensitivity and specificity can be selected in a variety of ways [22–25] based on the specific study goals.

Below, we describe two ways to obtain the genetic distance distributions of linked and unlinked infection pairs for a hypothetical pathogen with $R = 2$ and a substitution rate ($\mu$) of 1 substitution per genome per generation. We use the substitution rate rather than the pathogen mutation rate because our method concerns mutations *observed* between pathogen transmission events. We then use these genetic distance distributions to determine sensitivity and specificity, and ultimately to calculate the false discovery rate given a specific sample size and proportion. Here and henceforth, "generation" refers to a generation of transmission (not viral replication time).

## Empirical method

One way to estimate the relevant genetic distance distributions is to use existing data. Specifically, we need a subsample of infections for which sequencing data is available and we have a high degree of confidence—based on epidemiological data—of the true transmission relationships between included infections. For example, infected individuals who share a household versus community members with no known relationship. We can compute the genetic distance between every pair of pathogen sequences from this subsample and use the results to approximate the underlying genetic distance distributions between linked and unlinked infections in the population.

We illustrate this method on a simulated outbreak of approximately 1500 infections (data available at https://github.com/HopkinsIDD/phylosamplesize), created using the *outbreaker* R package [26,27] (see 'Outbreak simulations' below). To create our known subsample, we selected a small number of infections from early in the outbreak and extracted their true transmission links and simulated genomes. We then calculated the genetic distance matrix of sequences in this subsample and determined the genetic distance distributions for linked and unlinked infection pairs (Fig 2A). Next, we estimated the sensitivity and specificity at every mutation threshold (0 mutations, 1 mutation, etc.) and used the point closest to the (0,1) corner to determine the optimal threshold for differentiating between linked and unliked infections. In this case, the optimal threshold was 3 mutations, which had a sensitivity of 0.95 and a specificity of 0.88.

**Substitution rate method.** Observed pathogen substitution rates can also be used to estimate the genetic distance distributions, especially when a subsample of infections with known transmission histories is not available. If we assume that the number of mutations observed between two linked infections is Poisson distributed around the substitution rate and that we know the distribution of the number of generations between infections in the population, the probability of observing a specific genetic distance ($d$) between the sequences from any two

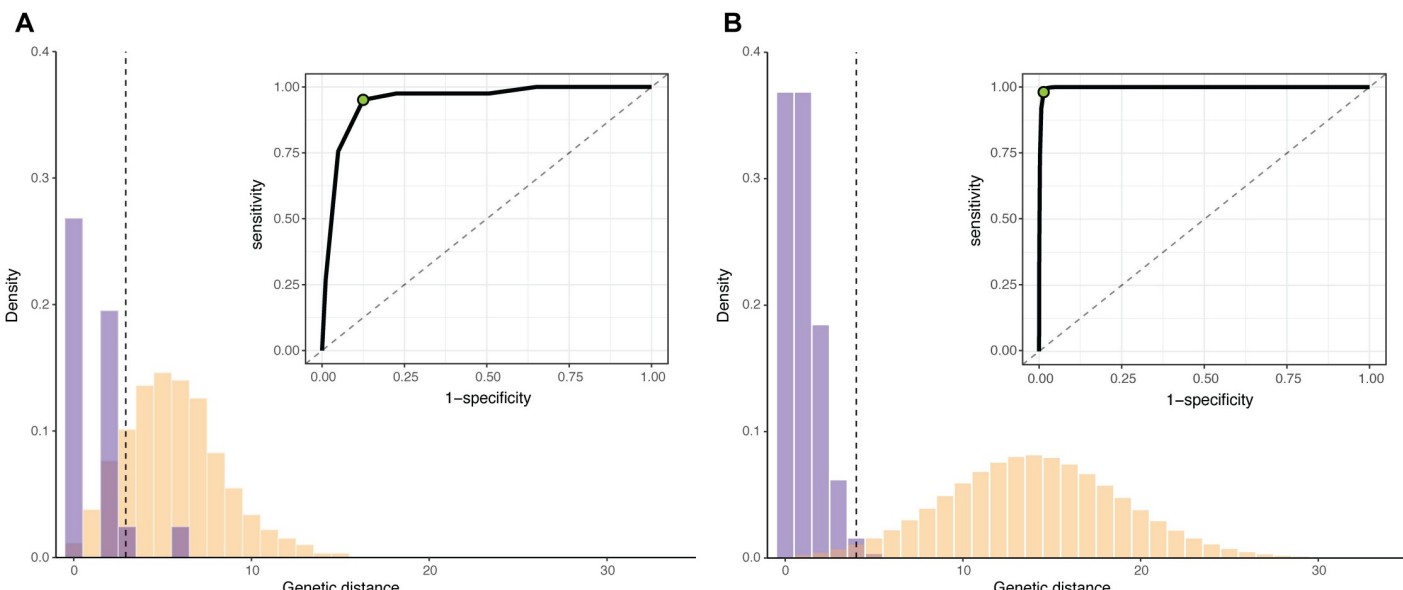

**Fig 2. Determining the sensitivity and specificity of a genetic distance threshold.** (**A**) Empirical distribution of genetic distances for linked (purple) and unlinked (yellow) infections for 50 infections selected from early in a simulated outbreak ($\mu = 1$ substitution/genome/generation, $R = 2$). Inset: receiver operating characteristic (ROC) for all possible genetic distance thresholds. Optimal threshold shown as green dot (ROC) and dashed vertical line (distribution). (**B**) Estimated distribution of genetic distances for linked and unlinked infections generated by the substitution rate method. Parameters and plots are as in (A).

infected individuals linked by transmission is:

$$\frac{1}{\sum_{i=1}^{g_{\text{link}}} g(i)} \sum_{i=1}^{g_{\text{link}}} g(i) \cdot f(d; i \cdot \mu) \tag{3}$$

where $g(i)$ is the probability of observing $i$ generations between infections, $g_{\text{link}}$ is the maximum number of generations between infections considered linked, $f(d;i\cdot\mu)$ is the probability of observing $d$ mutations between two infections separated by $i$ generations, and $\mu$ is the substitution rate per genome per generation (see **S2 Text**).

Similarly, the probability of observing a genetic distance $d$ between two infections not linked by transmission is:

$$\frac{1}{\sum_{i=g_{\text{link}}+1}^{g_{\text{max}}} g(i)} \sum_{i=g_{\text{link}}+1}^{g_{\text{max}}} g(i) \cdot f(d; i \cdot \mu) \tag{4}$$

Where $g_{\text{max}}$ is the maximum number of generations considered.

Since we assume that the number of substitutions between two linked infections is Poisson distributed, $f(d;i\cdot\mu)$ is simply the probability density function of a Poisson distribution with mean $i\times\mu$. Determining the distribution of generations between infections, however, is a non-trivial task [28–30] and depends on several factors, including the shape of the epidemic and the period of time from which infections are sampled (**S3 Fig**). In the examples included herein, we use simulations to empirically approximate this distribution (see **S2 Text**), but it is likely that adequate approximations can be obtained by other means—or that more sophisticated approaches can be employed to directly estimate the necessary genetic distance distributions [31].

Given the approximate generation distribution between infections, we calculated the genetic distance distributions for linked and unlinked infections for the pathogen described above. The optimal genetic distance threshold for distinguishing between linked and unlinked infections was 4 mutations (sensitivity = 0.98, specificity = 0.99) (**Fig 2B**). The empirical and substitution rate methods result in a similar, but not identical, optimal threshold for the pathogen in this example, likely due to sparse sampling in the empirical case.

Regardless of which method we choose, we can use the sensitivity and specificity values to calculate the probability of correctly identifying a true transmission pair ($\phi$) for this pathogen. We use **Eq 1**, allowing for each infection to have multiple transmission partners. We will also assume that we are able to sample 50% of the cases in this hypothetical outbreak of 1500 infections:

$$\phi = \frac{\eta\rho(R_{\text{pop}}+1)}{\eta\rho(R_{\text{pop}}+1) + (1-\chi)(M - \rho(R_{\text{pop}}+1) - 1)}$$

$$= \frac{0.98 * 0.5 * (1+1)}{0.98 * 0.5 * (1+1) + (1-0.99)(750 - 0.5 * (1+1) - 1)} = 0.116$$

We note that, despite a reproductive number (R) of 2, a single introduction into this outbreak means we should use $R_{\text{pop}} = 1$. Given our assumptions, we find that under 12% of our inferred linked infections—using a genetic distance threshold of 4 mutations—are likely to reflect true transmission relationships. A better specificity value is needed to achieve more confidence in direct transmission links, which can occur for pathogens that incur a significant number of mutations between infections considered linked [32]. For pathogens that do not

meet these criteria (as in the example here), it may not be possible to use genetic distance alone to distinguish between linked and unlinked infections (S4 Fig).

## Outbreak simulations

We used outbreak simulations to validate our approach. We simulated outbreaks using the 'simOutbreak' function implemented in the *outbreaker* R package [26]. For all simulations we assumed a large number of susceptible individuals in the population (n.hosts = 100,000), a genome length of 1,000 nucleotides, and no importation events (single source outbreak). We also assumed every infected individual transmitted their infection exactly one time step after infection, and ran the simulation for the number of generations needed to achieve a final outbreak size of approximately 1,000 infections (ln(1000)/ln(R)). We discarded simulations with an outbreak size of less than 100 or more than 2000 infected individuals; these discarded simulations did not count towards the total number of simulations for a given set of parameters. After simulating the source population, we randomly selected a predetermined proportion of infections from that population.

For each sampling proportion, we simulated outbreaks over a variety of substitution rates and reproductive numbers. We allowed the substitution rate to vary between 0.0001–4 mutations per genome per generation, and allowed the reproductive number to vary between 1.3–18. We chose these ranges to encompass substitution rates [33,34] and reproductive numbers [35] observed in actual human pathogens, and set the transition rate to be equal to the transversion rate for the purposes for this simulation. We note that, while pathogens can have reproductive numbers below 1.3, this was the minimum value that produced enough outbreaks with greater than 100 individuals in a reasonable amount of time. We divided each parameter range into 100 discrete values and ran simulations with all combinations of substitution rate and reproductive number, for a total of 10,000 simulations for each sampling proportion. We required simulated outbreaks to contain at least 100 and no more than 2000 infections for analysis. Validation plots were made in R using ggplot2 [36], and smoothed conditional means were calculated with the geom_smooth function from this package.

## Implementation

Functions for calculating the false discovery rate for a specific sample size or proportion are implemented in the R package *phylosamp*, freely available at: https://github.com/HopkinsIDD/phylosamp. This package also includes functions for calculating the necessary sample size based on a desired false discovery rate (inverse of **Eqs 1 and 2**), and functions to estimate the number of transmission pairs that will be observed given a sample size and a set of assumptions (e.g., multiple links and multiple transmissions, single link and single transmission, etc.). We also provide generation distributions for values of *R* between 1.3–18, derived from the simulations described in **S2 Text**.

## Applications to existing datasets

We used the phylosamp package to apply our method to an existing mumps virus dataset. We converted the reported substitution rate of $4.76 \times 10^{-4}$ substitutions/site/year [37] to 0.36 substitutions/genome/generation as follows:

$$\frac{4.76 \times 10^{-4} \text{substitutions}}{\text{site} \cdot \text{year}} \times \frac{15384 \text{ sites}}{\text{genome}} \times \frac{1 \text{ year}}{365 \text{ days}} \times \frac{18 \text{ days}}{\text{generation}}$$
$$= 0.36 \text{ subs/genome/generation}$$

We used a sampling proportion of 0.93, which is the fraction of samples from patients affiliated with Harvard University (71) that resulted in complete genomes. We also noted that the original mumps manuscript reports multiple lineages circulating within Harvard University, which would reduce the average reproductive number ($R_{pop}$) used to calculate the true discovery rate. However, decreasing this value again only decreases confidence in identified links, so we used $R_{pop} = 1$ to again calculate the upper bound of this estimate.

When applying the methods to a hypothetical SARS-CoV-2 outbreak, we converted a substitution rate of 24.896 substitutions/genome/year [38–40] to 0.34 substitutions/genome/generation using a generation time of 5 days [41]. The samplesize function in the phylosamp package gave the following error message when used with the optimum sensitivity and specificity (along with an outbreak size of 120 and true discovery rate of 0.9), indicating no amount of sampling would lead to high confidence in identified links: "Input values do no produce a viable solution."

## Results

### Method performance with known sensitivity and specificity

We used simulated outbreaks to validate the relationship between sample size and false discovery rate using genetic distance as our linkage criteria. We subsampled each outbreak and, using the known transmission relationships and genetic distances between simulated infections, calculated the false discovery rate at each possible genetic distance threshold in the subsample ("simulated FDR"). For each simulation (before subsampling), we also calculated the actual specificity and sensitivity at every relevant genetic distance threshold. We used these values and the observed $R_{pop}$ (roughly equal to one in most simulations) to then calculate the theoretical false discovery rate at a particular sampling proportion using **Eq 1**. We find that the theoretical false discovery rate is consistent with the simulated value for a wide array of pathogen substitution rates and reproductive numbers (**Fig 3**).

Overall, the bias of our estimate of the false discovery rate approached zero for all sampling proportions (**Table 2**). The average error was less than 0.04 in each case (i.e., false discovery rate estimate is off by no more than 4%), decreasing significantly with increased sample size or proportion sampled (**Tables 3** and **S1**). We note that special care should be taken with low sample sizes and low theoretical false discovery rates, as error rates can be particularly high in this range. Additionally, while our method is an unbiased estimator and overall correct in expectation, it is always possible for performance in a particular set of individuals sampled from a population to deviate substantially from expectation. As an example, in a small fraction of simulations, there were by chance no true transmission links (or, in some cases, no false positives) in our subsample. This fixes the simulated false discovery rate at 1 (or 0, when there are no false positives), which may not be representative of the overall relationship between sample size and false discovery rate and highlights how the specific infections sampled can affect results, particularly when sample sizes are low.

To better understand why the error rate of our estimator increases as the false discovery rate decreases, we stratified the simulation data by the sensitivity and specificity given a particular genetic distance threshold. We found that the error is highest when sensitivity is low and specificity is high (**S5A and S5B Fig**), which occurs when a high genetic distance threshold is used. This combination often produces low false discovery rates, but is highly dependent on sampling (namely, if any true positives or false positives are sampled). This leads to highly variable simulated false discovery rates and consequently higher error rates. Unsurprisingly, this analysis also highlights that a discrete threshold like genetic distance produces a limited number of possible sensitivity and specificity combinations (**S5C** and **S5D Fig**). Therefore,

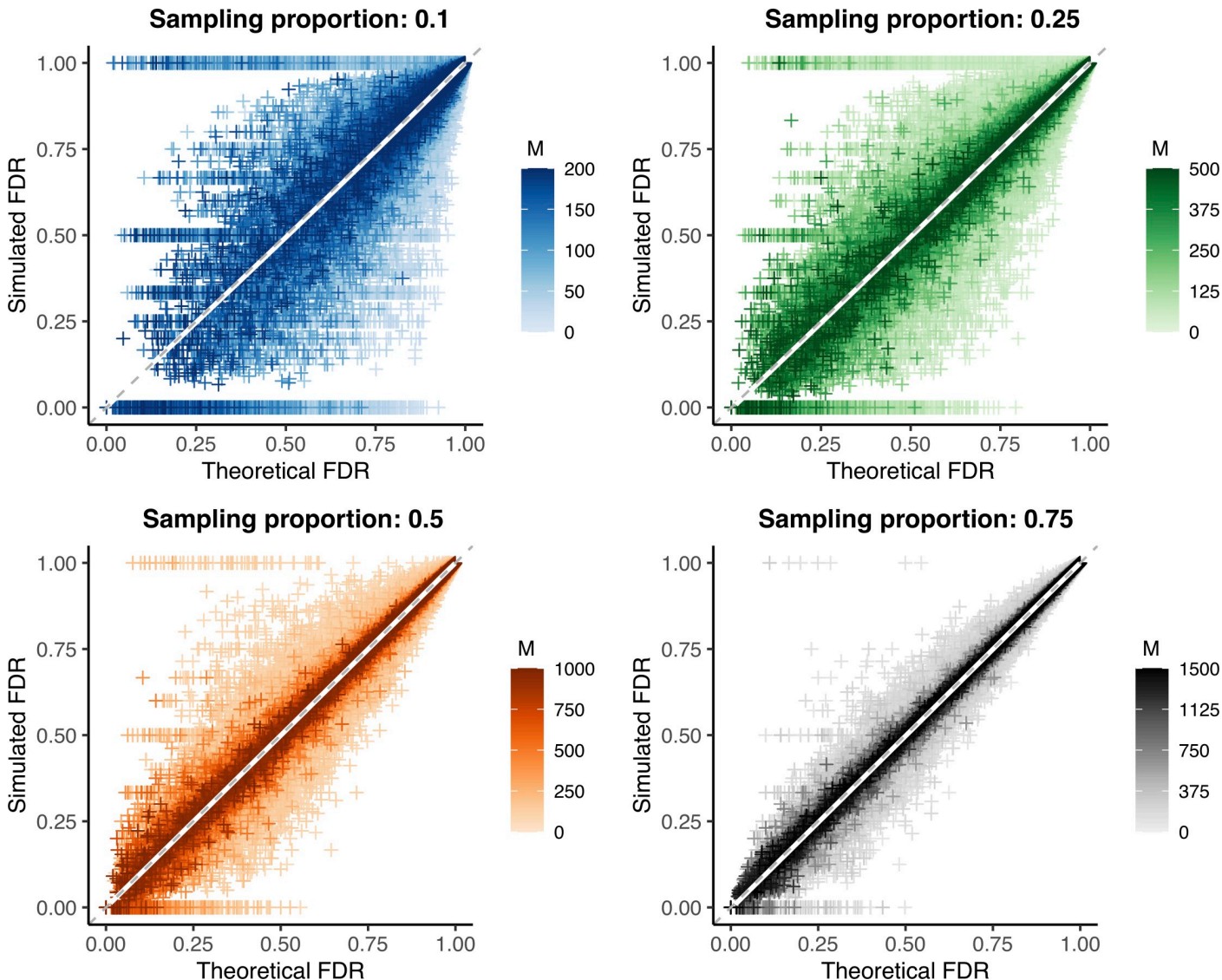

**Fig 3. Predicted versus observed false discovery rate in outbreak simulations.** Theoretical versus simulated false discovery rate (FDR) for each genetic distance threshold in 10,000 simulations of varying substitution rate and reproductive number (approximately 260,000 points per plot, see **Tables 2 and 3**). Outbreak sizes range from 100–2000, as described in **Methods**. White line: smoothed conditional mean; grey dashed line: y = x line. Increasing values of the sample size ($M$) are plotted in darker color; because the maximum outbreak size is fixed at 2000, the maximum sample size differs for each sampling proportion. Increasing both the sample size and proportion reduces bias and error, see **Tables 2 and 3**.

**Table 2. Bias of calculated false discovery rate for simulations with fixed sampling proportion.**

| | $\rho = 0.10$ | $\rho = 0.25$ | $\rho = 0.50$ | $\rho = 0.75$ | All $\rho$ values | N |
|---|---|---|---|---|---|---|
| FDR = 0.00–0.25 | -0.0006 | 0.0045 | 0.0001 | 0.0036 | **0.0022** | 17,900 |
| FDR = 0.25–0.50 | 0.0044 | 0.0045 | 0.0009 | 0.0032 | **0.0032** | 31,633 |
| FDR = 0.50–0.75 | 0.0064 | 0.0039 | 0.0006 | 0.001 | **0.0029** | 51,069 |
| FDR = 0.75–1.00 | 0.0001 | 0.0001 | <0.0001 | <0.0001 | **0.0001** | 965,125 |
| All FDR Values | **0.0005** | **0.0005** | **0.0001** | **0.0002** | **0.0003** | 1,065,727 |
| N | 261,360 | 267,239 | 268,900 | 268,228 | 1,065,727 | |

**Table 3. Error of calculated false discovery rate for simulations with fixed sampling proportion.**

| | $\rho = 0.10$ | $\rho = 0.25$ | $\rho = 0.50$ | $\rho = 0.75$ | All $\rho$ values | N |
|---|---|---|---|---|---|---|
| FDR = 0.00–0.25 | 0.2135 | 0.1359 | 0.0799 | 0.0401 | **0.098** | 17,900 |
| FDR = 0.25–0.50 | 0.2751 | 0.1583 | 0.079 | 0.0416 | **0.1275** | 31,633 |
| FDR = 0.50–0.75 | 0.2057 | 0.0979 | 0.0478 | 0.0259 | **0.092** | 51,069 |
| FDR = 0.75–1.00 | 0.0155 | 0.0069 | 0.0035 | 0.002 | **0.007** | 965,125 |
| All FDR Values | **0.032** | **0.0181** | **0.0097** | **0.0052** | **0.0161** | 1,065,727 |
| N | 261,360 | 267,239 | 268,900 | 268,228 | 1,065,727 | |

obtaining reasonable estimates for these values in tandem is of key importance when using our method to estimate the false discovery rate of a phylogenetic study.

## Method performance with estimated sensitivity and specificity

We repeated the false discovery rate comparison described above, but instead of using the actual sensitivity and specificity observed in each simulation, we calculated these parameters from the substitution rate used to generate that simulated outbreak (**Fig 4**). To reduce reliance on simulation data to calculate necessary parameters, we used $R_{pop} = 1$ rather than the empirical value.

Under this more realistic set of assumptions, we observe a slight bias, though overall values remain less than one percent (**S2 and S3 Tables**). However, while mean bias is very low on average, it is greater when the theoretical false discovery rate is low, reaching an average of nearly 8% off the simulated value for predicted false discovery rates less than 25%. Average error rates were similarly slightly increased, but remained less than 4% overall. Despite these trends, the vast majority of false discovery rate estimates (as well as sensitivity and specificity estimates) fall very close to their true values (**Fig 5**). This observation holds true when only examining the optimal genetic distance threshold (using the closest to the (0,1) corner method, as described in **Methods**) (**S6 Fig**) rather than estimated values at all thresholds shown in **Figs 4 and 5**.

Given that correct sensitivity and specificity values are an important component of calculating the theoretical false discovery rate, we looked at the specific estimates for these parameters generated by our substitution rate method. When considering only direct transmissions as linked (as we do throughout these simulations), **Eq 3** simplifies to simply a Poisson distribution around the substitution rate, resulting in highly accurate and precise sensitivity estimates (**Figs 5** and **S7**). However, we find that our estimates for specificity have a positive bias regardless of sample size or proportion (**Figs 5** and **S8 and S9**). We hypothesized that inaccuracies in the estimated specificity cause the bias observed in the false discovery rate estimate and were due to the distribution of generations between infections used in our calculation; as discussed in **Methods**, this is a non-trivial distribution that we estimated by averaging over many simulations (see **S2 Text** for details).

To test this hypothesis, we used the actual distribution of generations between infections from each simulation in our calculation of specificity (sensitivity estimates are unaffected by this distribution when considering only direct transmissions, as described above). We find that this does in fact reduce bias in our specificity estimates (**Fig 6**) and leads to largely unbiased (<2%) estimates of the false discovery rate, even at low theoretical false discovery rate values (**S10 Fig** and **S4 Table**).

## Application of the sampling framework

**Illustrative retrospective example.** To illustrate our sample size calculation framework, we used a publicly available dataset from an outbreak caused by a well characterized pathogen

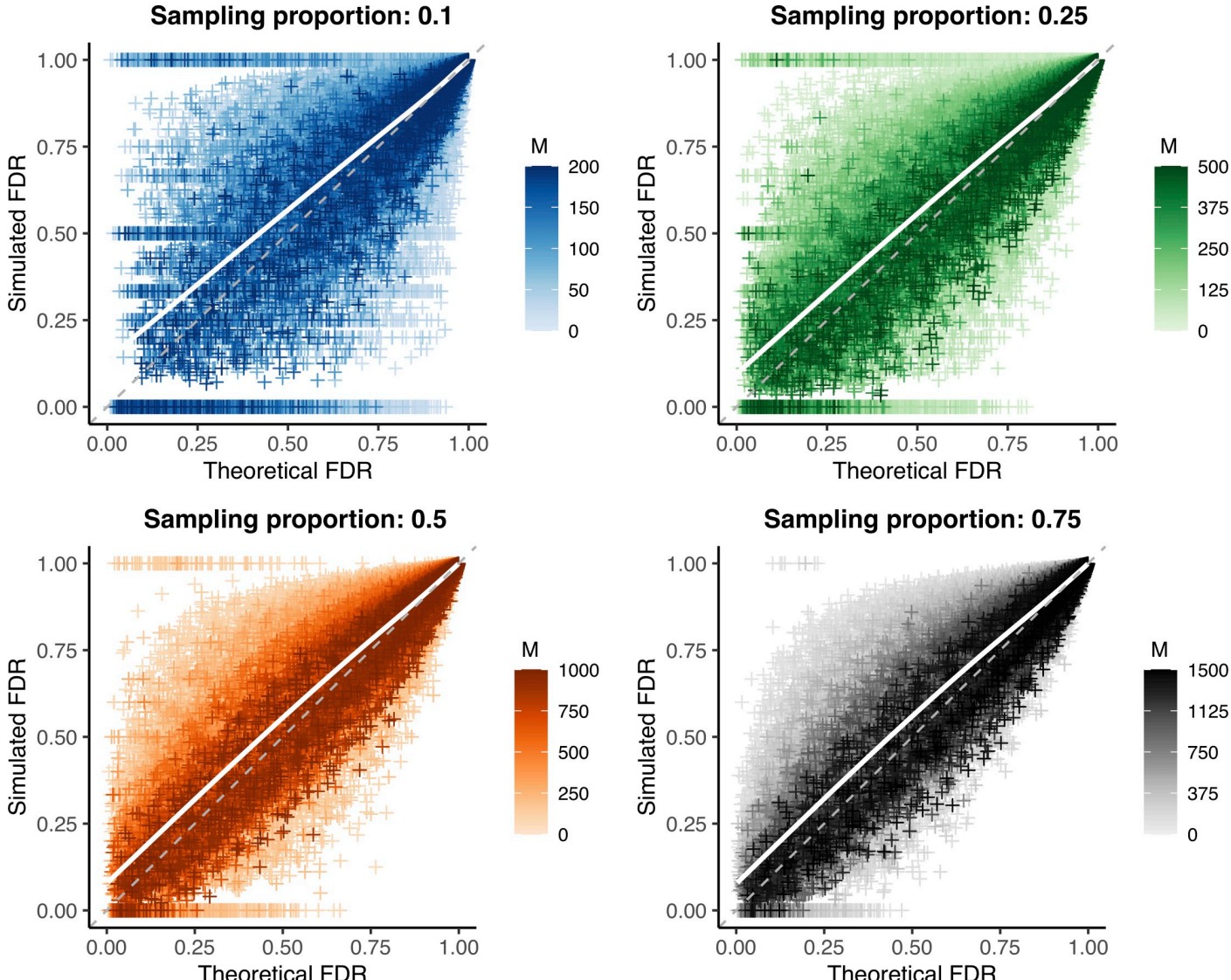

**Fig 4. Validation of substitution rate method to calculate sensitivity and specificity.** Theoretical versus simulated false discovery rate (FDR) for each genetic distance threshold in 10,000 simulations of varying substitution rate and reproductive number (approximately 260,000 points per plot, see **Tables 2 and 3**). Outbreak sizes range from 100–2000, as described in **Methods**. White line: smoothed conditional mean; grey dashed line: $y = x$ line. Increasing values of the sample size ($M$) are plotted in darker color; increasing both the sample size and proportion reduces bias and error, see **S2** and **S3 Tables**.

(mumps virus) that had been subject to both genomic and epidemiological analysis [37]. We first used the substitution rate method described above to calculate the sensitivity and specificity of genetic distance as a linkage criteria using the substitution rate reported in the study (molecular clock rate = $4.76 \times 10^{-4}$ substitutions per site per year). We converted this substitution rate to 0.36 substitutions per genome per generation using the mean generation interval estimated in the study (18 days), which falls within previous estimates of this parameter [42–44]. We used the effective reproductive number reported for Harvard University (1.70) to estimate the generation time distribution using our *phylosamp* package, as shown in the R code below:

```
library(phylosamp)
data("gen_dist_sim")
```

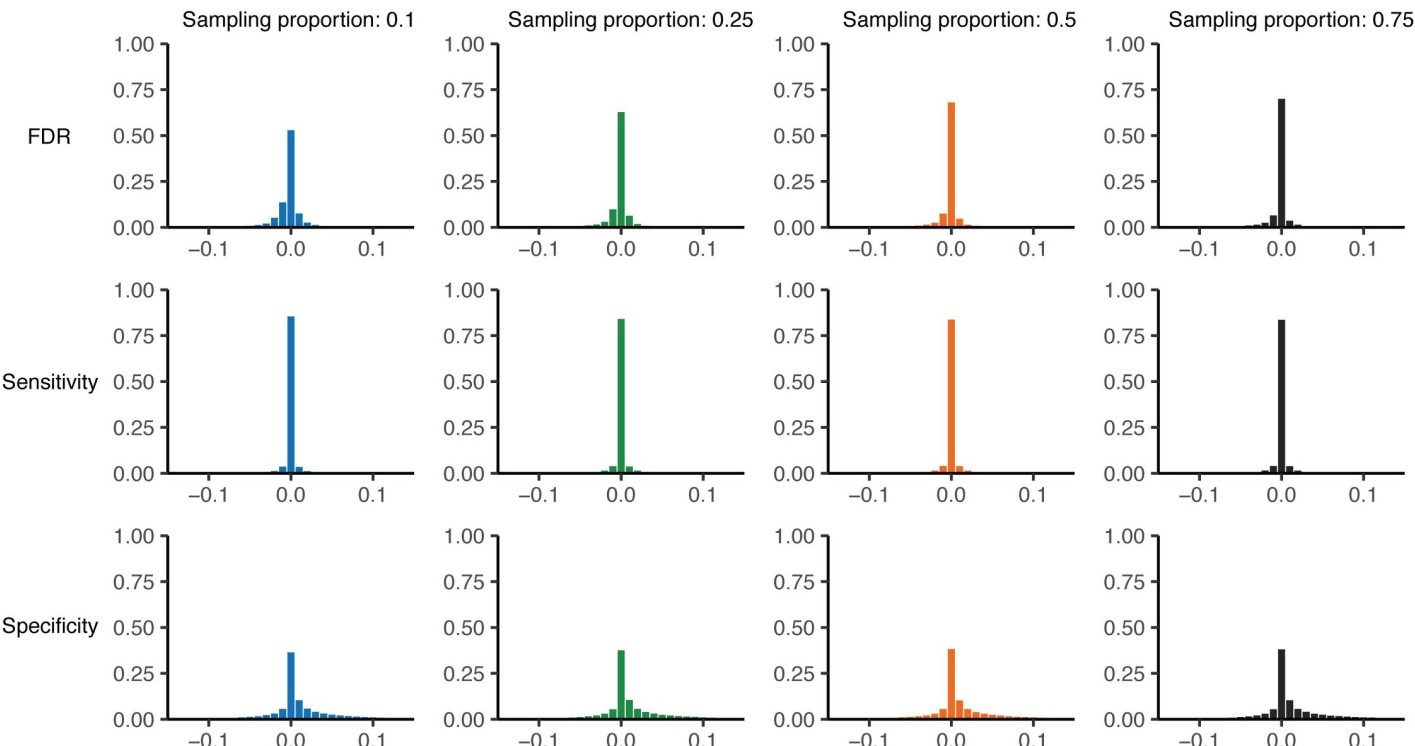

**Fig 5. Histogram of raw parameter error using substitution rate method.** Theoretical minus simulated parameter values for each genetic distance threshold in 10,000 simulations of varying substitution rate and reproductive number for a given sampling proportion (see **Fig 4**). Top row: theoretical minus simulated false discovery rate; middle row: theoretical minus simulated sensitivity; bottom row: theoretical minus simulated specificity. Colors correspond to sampling proportion as in **Fig 4**.

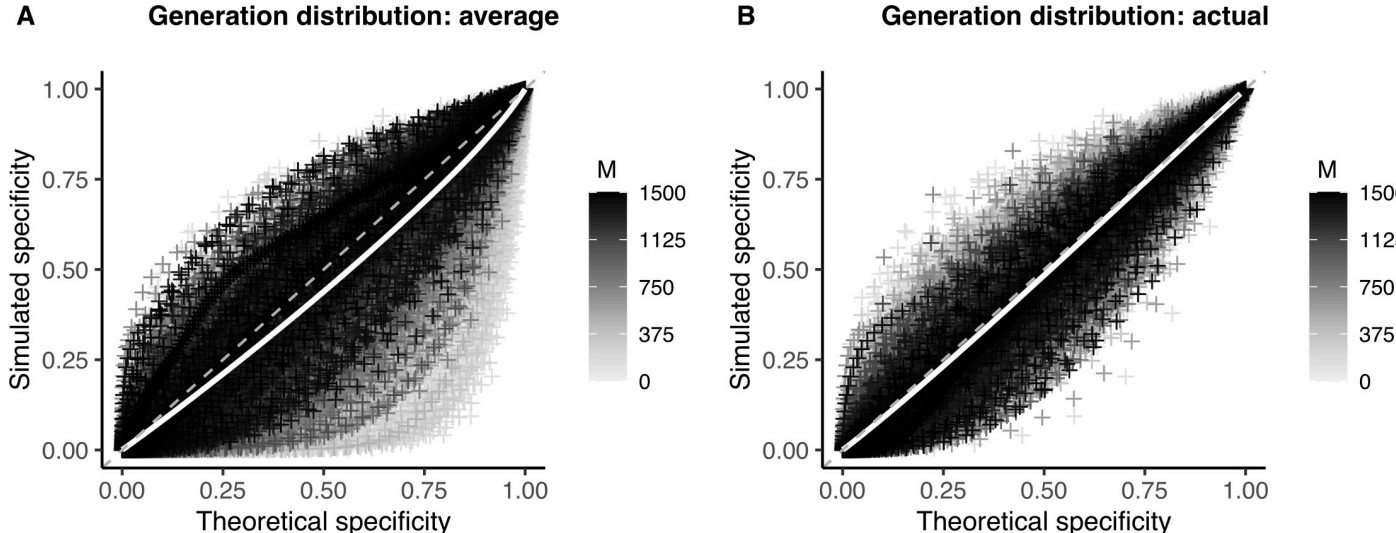

**Fig 6. Effect of the generation distribution on specificity of the linkage criteria.** Theoretical versus simulated specificity for each genetic distance threshold in 10,000 simulations of varying substitution rate and reproductive number (proportion sampled = 0.75). White line: smoothed conditional mean; grey dashed line: y = x line. Increasing values of the sample size (*M*) are plotted in darker color. (**A**) Theoretical sensitivity and specificity calculated using average distribution of generations between infections from simulations (see **S2 Text**). (**B**) Theoretical sensitivity and specificity calculated using the actual distribution of generations between infections from that simulated outbreak.

```
mgd <- as.numeric(gen_dist_sim[gen_dist_sim$R = = 1.70, -(1:2)])
get_optim_roc(sens_spec_roc(cutoff = 1:20,mut_rate = 0.36,
  mean_gens_pdf = mgd))
```

This method results in an optimal sensitivity of 0.95 and specificity of 0.95 using a cutoff of two mutations.

We then used these parameter values to calculate the true discovery rate of our linkage criteria, i.e., the proportion of identified links (whole mumps genomes differing by <2 mutations) that represent actual transmission pairs. We focused on the part of the mumps outbreak within Harvard university, for which 66 whole genomes sequences were generated from 71 unique patient samples. While the true number of cases at Harvard was likely significantly higher, this provides a maximum sampling proportion of 93% of infections. Using the *phylosamp* package, we calculated the true discovery rate as follows:

```
truediscoveryrate(eta = optim$sensitivity,chi = 1-optim$specificity,
  rho = 0.93,M = 66,R = 1)
```

Using our method, we calculated a true discovery rate of 0.35. This low value suggests that genetic distance alone would not be sufficient to identify specific transmission links within the Harvard community during this mumps outbreak. This is in line with the findings of the original paper, which demonstrates the need for both genomic and epidemiological data to understand transmission, and emphasizes the frequent need for such epidemiological data to achieve the required specificity for high confidence estimation of transmissiosn links.

**Illustrative prospective example.**   To demonstrate how our method could be used to estimate the sample size needed to identify transmission links with 90% confidence (i.e, a true discovery rate of 0.9), we applied our method to a hypothetical COVID-19 outbreak in an unvaccinated community with 120 infections. We calculated the sensitivity and specificity of genetic distance using a substitution rate of 0.34 mutations per genome per generation [38–41] and an *R* value of 3, consistent with many efforts [45,46]:

```
mgd <- as.numeric(gen_dist_sim[gen_dist_sim$R = = 3, -(1:2)])
get_optim_roc(sens_spec_roc(cutoff = 1:20,mut_rate = 0.34,
  mean_gens_pdf = mgd))
```

This method results in an optimal sensitivity of 0.95 and a specificity of 0.84 using a cutoff of two mutations. Using these parameters, we found that not even perfect sampling could lead to a true discovery rate of at least 0.9:

```
samplesize(eta = optim$sensitivity,chi = 1-optim$specificity,N = 120,
  R = 1,phi = 0.9)
```

This suggests that genetic distance alone is not sufficient to differentiate linked and unliked SARS-CoV-2 infections at high confidence. However, if we could identify additional phylogenetic or epidemiological criteria that would increase the specificity to 0.999 (keeping the sensitivity at 0.95), a sample size of 11 would achieve our desired confidence in direct transmission links. Additionally, it may be more fruitful to focus on cases linked within several generations of transmission, during which additional mutations would have time to accumulate.

## Discussion

We have developed a mathematical framework for making informed sampling decisions in pathogen genome sequencing studies. Specifically, this framework allows for easy calculation of the relationship between the number or proportion of infections sampled during an outbreak and the ability of some phylogenetic or epidemiological criteria to correctly identify infections within this sample that are linked by direct transmission. Understanding this relationship is crucial to making correct inferences about pathogen transmission patterns, especially as genomic studies are becoming more feasible and widely used to answer both scientific and public health questions.

This framework is broadly applicable to a variety of phylogenetic or epidemiological approaches, as long as the sensitivity and specificity of the criteria can be approximated. With a basic understanding of the pathogen and the criteria being used, researchers can more effectively design studies that correctly identify transmission pairs with a known level of confidence. Additionally, this generalizable method (available as a free software, the R package *phylosamp*) provides a metric by which reviewers of these studies can evaluate their conclusions. We apply our method to simulated outbreaks using genetic distance as the linkage criteria and find that we can effectively estimate the false discovery rate for a variety of pathogen substitution rates, reproductive numbers, and relevant genetic distance thresholds. It is important to note, however, that for a given sensitivity and specificity, there may not always be a study design that achieves the desired false discovery rate.

Performance of the method presented depends on our ability to estimate the sensitivity and specificity of a particular linkage criteria. While we present two methods for doing this—empirically and theoretically using the substitution rate of the pathogen—implementing either in practice is not without challenges, and improved estimation of these values may be a fruitful area for future research. For instance, the substitution rate based approach also depends on the distribution of the number of generations of transmission between infections in the underlying population. Although distributions derived from simulations (provided as part of the *phylosamp* package) provide a reasonable proxy, estimates of sensitivity and specificity are much improved when using the exact generation distribution, which currently can only be determined from complete knowledge of all transmission events. Further research into all the factors affecting this distribution will be necessary to improve its estimation. Likewise, there are challenges to the empirical approach, particularly for novel pathogens.

Better performance can likely be obtained by not restricting ourselves to genetic distance alone when determining a linkage criteria. Genetic distance is easy to determine from sequence data, but this simple metric does not take into account ancestral relationships or uncertainty around these relationships, and is limited to discrete mutational changes. Applying more complex phylogenetic criteria may allow us to learn more about transmission relationships, though there is a limit to the extent to which genetic data can be used to distinguish infections in fast-spreading (or slow-mutating) pathogen outbreaks. There are several examples of outbreaks in which multiple infected individuals have the same consensus viral genome [32]. In this case, incorporating epidemiological data (e.g., location, time of symptom onset) may be important in determining which infections are unlikely to be linked. This incorporation of additional data may complicate calculation of the sensitivity and specificity, so developing the methodology around calculating these parameters will be important to further development of our method. This will likely build on a larger effort to better integrate epidemiological and genomic data into pathogen transmission studies [26,47–49].

The application of our methodology to a previous mumps outbreak and a hypothetical COVID-19 outbreak highlights the need to move beyond genetic distance as a linkage criteria; for pathogens with a substitution rate similar to that of mumps virus, genetic distance is not enough to differentiate between linked and unlinked cases even in densely sampled outbreaks. In trying to apply this method to other outbreaks, it also became clear that well-characterized substitution rates and reproductive numbers are essential for calculating sensitivity and specificity using our method, and that these parameters are less clearly defined for pathogens with long and variable generation times, such as bacterial infections. Variable periods of replication within a host makes it difficult to characterize a per-generation substitution rate that is broadly applicable over the entire outbreak and can be used to estimate sensitivity and specificity. In these cases, more nuanced criteria such as phylogenetic relatedness will likely be more informative than the number of mutations between sequenced infections; while we provide

instructions for using genetic distance as a linkage criteria in order to give a concrete example of calculating sensitivity and specificity, the primary focus of this manuscript is to demonstrate how they can be used to calculate or evaluate sample sizes.

While in this manuscript we have focused on direct transmission pairs, our framework is designed to be extensible to alternative definitions of linkage; for example, infections connected within a specified number of transmission events. Expanding the definition of linkage to include such indirect transmissions has a number of useful applications in outbreak research, such as identifying and connecting transmission clusters. This method could also be extended to more complex direct transmission relationships, for example when within-host evolution results in the existence of viral quasispecies within infected individuals, each of which has some potential of being transmitted. In all of these scenarios, it is equally important to understand the sample size needed to make the desired inferences.

We hope that this work represents a step towards developing a larger theory of study design for making inferences from pathogen sequence data, but recognize it is only a step. The focus of this paper is sample size and the impact of undersampling, but spatial and/or temporal biases are also important for determining which infections are sampled [50–52]. For example, understanding routes of direct transmission may require dense sampling of a small group of highly-connected individuals, while understanding general transmission trends over the course of a geographically-dispersed outbreak may require us to sample broadly over space and time. Additionally, it will be important to take into account the contact network underlying pathogen transmission, since some individuals may be more likely to transmit their infection to others. Finally, the goal of linking infections is seldom the linkages themselves, but the larger inferences about risk and transmission derived from those linkages. Adapting the techniques here to more directly link sample size calculations to these outcomes is an important next step.

## Supporting information

**S1 Fig. Sample size and false discovery rate given single linkage and single transmission.** (**A**) Effect of sample size (red lines) or proportion sampled (blue lines) on the expected number of linked pairs (upper plots) or the false discovery rate of linked pairs (lower plots). The specificity and sensitivity are held constant. (**B**) Effect of varying the sensitivity and specificity of the linkage criteria on the false discovery rate (FDR).
(TIF)

**S2 Fig. Estimating the average reproductive number in a population.** Two hypothetical outbreaks with a pathogen reproductive number ($R$) equal to 2 and a total of 15 infections. Black circles represent infections; blue circles represent infections who have not yet infected others, or whose descendents are outside the sampling frame. (**A**) Outbreak caused by a single introduction, meaning there were 14 transmission events and 15 total infections. In other words, $R_{pop} = \frac{14}{15} = 0.933$. (**B**) Outbreak caused by two separate introductions, meaning there were only 13 infection events in the sampling frame, resulting in $R_{pop} = \frac{13}{15} = 0.867$.
(TIF)

**S3 Fig. Effects of R and G on the distribution of generations between cases.** Distribution of the number of generations between infections averaged over 1000 simulated outbreaks with reproduction number R and number of generations of transmission G. Distributions are shown for three values of R (rows). Left column: distribution of generations between infections after 3 generations of transmission; middle column: distribution after $ln(1000)/ln(R)$ generations of transmission (see Methods); right column: distribution after $ln(1000)/ln(R)+2$

generations of transmission.
(TIF)

**S4 Fig. Genetic distance distributions for different types of pathogens.** (**A**) Distribution of genetic distances for linked (purple) and unlinked (yellow) infections for a hypothetical pathogen with substitution rate = 1 substitution/genome/generation and $R$ = 1.5. Inset: receiver operating characteristic (ROC) curve for all possible genetic distance cutoff values. Optimal threshold shown as green dot (ROC) and dashed vertical line (distribution). (**B**) Distribution of genetic distances for linked and unlinked cases for a hypothetical pathogen with substitution rate = 0.2 mutations/genome/generation and $R$ = 3. Inset: ROC curve for all possible genetic distance cutoff values for this pathogen. The optimal threshold is shown as in (A).
(TIF)

**S5 Fig. Error of false discovery rate calculation by sensitivity and specificity.** (**A**) Average false discovery from 10,000 simulated outbreaks (proportion sampled = 0.75) binned by sensitivity and specificity (bin size = 0.02). Grey = no genetic distance thresholds in simulation produced this combination of sensitivity and specificity. (**B**) Zoom view of (A), with specificity ranging from 0.9–1 (bin size = 0.002). (**C**) Number of data points with sensitivity and specificity in the desired bins (i.e., number of data points used to calculate average error in panel (A). (**D**) Zoom view of (C), with specificity ranging from 0.9–1.
(TIF)

**S6 Fig. Histogram of raw parameter error using substitution rate method (optimal threshold only).** Theoretical minus simulated parameter values for the optimal genetic distance threshold (determined by selecting the threshold for which the point at (1-specificity, sensitivity) is closest to the (0,1) corner) in 10,000 simulations of varying substitution rate and reproductive number for a given sampling proportion. Top row: theoretical minus simulated false discovery rate; middle row: theoretical minus simulated sensitivity; bottom row: theoretical minus simulated specificity. Colors correspond to sampling proportion as in **Fig 4**.
(TIF)

**S7 Fig. Predicted versus observed sensitivity using substitution rate method.** Theoretical versus simulated sensitivity for each genetic distance threshold in 10,000 simulations of varying substitution rate and reproductive number. White line: smoothed conditional mean; grey dashed line: $y = x$ line. Increasing values of the sample size ($M$) are plotted in darker color.
(TIF)

**S8 Fig. Predicted versus observed specificity using substitution rate method.** Theoretical versus simulated specificity for each genetic distance threshold in 10,000 simulations of varying substitution rate and reproductive number. Outbreak sizes range from 100–2000, as described in **Methods**. White line: smoothed conditional mean; grey dashed line: y = x line. Increasing values of the sample size ($M$) are plotted in darker color.
(TIF)

**S9 Fig. Histogram of raw specificity error using substitution rate method by sample size and proportion.** Theoretical minus simulated specificity for each genetic distance threshold in 10,000 simulations of varying substitution rate and reproductive number for a given sampling proportion. Each column represents 10,000 simulations with a specific sampling proportion (colors as in **Fig 4**) and sample size within each proportion (determined by the final outbreak size) goes from low (top row) to high (bottom row).
(TIF)

**S10 Fig. Predicted versus observed false discovery rate using actual generation distribution.** Theoretical versus simulated false discovery rate (FDR) for each genetic distance threshold in 10,000 simulations of varying substitution rate and reproductive number. Theoretical FDR is calculated using the actual distribution of generations between infections from the corresponding simulated outbreak. White line: smoothed conditional mean; grey dashed line: $y = x$ line. Increasing values of the sample size ($M$) are plotted in darker color.
(TIF)

**S1 Table. Error of false discovery rate calculation by sample size.**
(PDF)

**S2 Table. Bias and error of false discovery rate calculation using substitution rate method.**
(PDF)

**S3 Table. Error and of false discovery rate calculation using substitution rate method by sample size.**
(PDF)

**S4 Table. Bias and error of false discovery rate using actual generation distribution.**
(PDF)

**S1 Text. Deriving probably of transmission given linkage.**
(PDF)

**S2 Text. Determining sensitivity and specificity of genetic distance as a linkage criteria.**
(PDF)

## Acknowledgments

We thank Stuart Ray for his insightful comments on the manuscript.

## Author Contributions

**Conceptualization:** Justin Lessler.

**Formal analysis:** Shirlee Wohl, John R. Giles, Justin Lessler.

**Funding acquisition:** Justin Lessler.

**Methodology:** Shirlee Wohl, John R. Giles, Justin Lessler.

**Software:** Shirlee Wohl, John R. Giles.

**Supervision:** Justin Lessler.

**Validation:** Shirlee Wohl.

**Visualization:** Shirlee Wohl, Justin Lessler.

**Writing – original draft:** Shirlee Wohl, Justin Lessler.

**Writing – review & editing:** Shirlee Wohl, John R. Giles, Justin Lessler.

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
