## [Decision Letter · Decision Letter 0]

20 Jan 2021

Dear Dr. Lessler,

Thank you very much for submitting your manuscript "Sample Size Calculation for Phylogenetic Case Linkage" for consideration at PLOS Computational Biology.

As with all papers reviewed by the journal, your manuscript was reviewed by members of the editorial board and by several independent reviewers. In light of the reviews (below this email), we would like to invite the resubmission of a significantly-revised version that takes into account the reviewers' comments.

As you'll see, the reviews are mixed. While Reviewer 2 has only minor suggestions for improvements in the clarity of the text, Reviewers 1 and 3 have more substantive comments. In particular, both would like to see the methods illustrated using an openly available real data set, and I strongly encourage the authors to do so. Reviewer 3 (who previously reviewed the manuscript at eLife) still has some substantive methodological concerns, although I think that these can be addressed.

We cannot make any decision about publication until we have seen the revised manuscript and your response to the reviewers' comments. Your revised manuscript is also likely to be sent to reviewers for further evaluation.

Sincerely,

Virginia E. Pitzer, Sc.D.

Deputy Editor-in-Chief

PLOS Computational Biology

Virginia Pitzer

Deputy Editor-in-Chief

PLOS Computational Biology

As you'll see, the reviews are mixed. While Reviewer 2 has only minor suggestions for improvements in the clarity of the text, Reviewers 1 and 3 have more substantive comments. In particular, both would like to see the methods illustrated using an openly available real data set, and I strongly encourage the authors to do so. Reviewer 3 (who previously reviewed the manuscript at eLife) still has some substantive methodological concerns, although I think that these can be addressed.

Reviewer's Responses to Questions

**Comments to the Authors:**

Reviewer #1: Wohl et al present a statistical framework for calculating sample sizes for robust determinations of the infector-infectee pairs within transmission chains of pathogen genomic epidemiology studies. Their framework also provides methods for calculating FDR and the expected number of true transmission pairs from the specificity and sensitivity of the linkage criteria (genetic distances), sample size, the proportion of samples sequenced, and the effective reproductive number of the pathogen analysed. The authors demonstrate the utility of this framework with simulation data and developed the R package “phylosamp” to provide an implementation of their framework.

This manuscript addresses a neglected problem in many genetic epidemiology studies regarding the level of sequencing required to be carried out in order for robust conclusions to be made when reconstructing transmission chains of pathogen outbreaks using WGS data. The work is novel as there are a lack of current formal agreed upon standards for carrying out this aspect study design, and is both relevant and timely given the increasing widespread adoption of genetic epidemiology techniques for understanding pathogen transmission dynamics. Further, the manuscript is well written, the underlying methodology well described, and the use cases of the software and limitations are appropriately discussed.

Please find my comments below, divided into different sections for (a) the manuscript describing the framework and (b) the R package phylosamp. I hope these are useful to the authors.

A. Manuscript comments:

(1) The reliance of phylosamp at present on genetic distances alone as the linkage criteria presents a key limitation in calculating appropriate sample sizes and other parameters for a study concerning slowly evolving pathogens where there is limited genetic variation accumulating between transmission pairs/generations which prohibits their detection from WGS alone. I recognise that the focus of this manuscript is a first step towards more comprehensive approaches, and that these concerns are discussed in both the manuscript, and in previous supplied reviews from a submission to eLife, but also believe that this limits the utility of the software for many genetic epidemiology studies.

(2) While the simulation data provide a useful and convincing illustration of the framework, it would be excellent to also see an example application of phylosamp to an existing published pathogen dataset to further demonstrate its utility. Again, I recognise that this has been discussed in previous reviews from a submission to eLife, but the inclusion of such data would present a substantial improvement to the work and encourage further adoption of the framework.

(3) The definition of Rpop provided from line 100, where it is first introduced requires rephrasing for clarity. While this is better described later in the manuscript from line 149, the earlier text could be clarified to avoid the reader having to scroll back and forth throughout the paper. I recognise that this text has already been refined based on the reviewer comments from the previous submission to eLife, however, it could benefit from further refinement for improved clarity and flow.

(4) Figure 1B: Does each white dot indicate the sensitivity and 1-specificty for a SNP/genetic distance increased in increments of 1? i.e. 0, 1, 2, 3, 4, … SNPs? If so, it would be helpful to indicate the values of these increments either by annotation of the figure itself or expansion of the figure legend to improve clarity.

(5) Line 242: There don’t appear to be any citations for the range of effective reproductive numbers of human pathogens explored in simulation studies.

(6) Figure S5: It appears that either the figure panels or the legend descriptions might be inverted for A and B, as well as C and D.

(7) The authors have put substantial effort into making their work openly available by submitting a preprint on medrxiv and providing all code and data files required to reproduce their analyses and manuscript figures via github (available at: https://github.com/HopkinsIDD/phylosamplesize). I was able to reproduce all figures and analyses until line #113 of figures.Rmd at which point I was unable to proceed further.

i.e.

# first time only: calculate tfdr from simulations and save to file

calc.tfdr(simdata="data/simdata_var_N10000",rho_values=c(0.1,0.25,0.5,0.75),max_sim_size=2000,

sens_spec_method="sim",mgd=mgd,outdir="data/full_data_sim.Rdata")

I think this might be due to the files being specified by the prefix “simdata_var_N10000” where it might need to be instead specified as “simdata_var_gen_N10000”, but the authors may need to look into this further.

B. Phylosamp R package and documentation comments:

Code from the R package was clearly structured and generally well commented. The package is freely available and easily installed via the devtools library. I was able to reproduce the results from the vignette code easily and without issue, and found the explanations very clear and informative. I have provided some comments on the R package and documentation below that I hope are useful to the authors, but do not regard any of these to be critical changes, nor do I require that these suggested changes be made for the publication of this manuscript.

- In the vignettes it may be worth providing a simple reiteration of what each argument provided to the function is in the vignette (e.g. for eta, chi, rho)

- There appears to be a typo at the top of the ‘Illustrated examples’ vignette page, I think “this vignette…” should perhaps be “In this vignette…”.

- When using the help operator in R, I found the package to be well documented for all functions, but at times it was a little unclear to me which defaults were used when these are not supplied explicitly by the user i.e. the assumption argument. I think based on the manuscript and example function provided via the help operation in R this is mtml for ‘multiple-transmission multiple-linkage’, but perhaps this could be further clarified in the package documentation

Reviewer #2: Wohl et al. present a method for understanding how sampling, both in terms of overall depth and in terms of proportion, influences how accurately we can identify true infector-infectee pairs (linked cases) from a phylogeny of pathogen genomes. This theoretical area of genomic epidemiology is sorely underdeveloped, especially when compared to the rigorous theoretical framework for sampling design available for traditional epidemiological studies. This work is the first real step I’ve seen to develop sample size calculations for genomic epidemiological studies. The manuscript is clearly written, and I am satisfied by how the authors have addressed previous reviewer comments. While this work should be accepted, I do have some minor comments that should be addressed to avoid reader confusion and position this paper in the appropriate context. These comments do not require further analytic work; they are only textual changes.

1. In the Introduction the authors draw on many examples of how pathogen genomic information can be used to investigate public health questions (lines 34-37) at multiple scales (lines 47-49), and declare that all of those questions can be boiled down to a question of asking whether pairs of infections are related. I disagree with this, especially within the context of sampling. Sampling considerations within phylogeographic studies, which seek to infer patterns of spatial linkage, center on the assumption that sampling must be sufficiently broad and random to have fully sampled all circulating genetic lineages, generally at an intensity that is proportional to a lineage’s prevalence. For those questions I don’t see how it’s important that linked pairs are captured, and thus I don’t see how this method would help me to design better phylogeographic studies. I would recommend that the authors pivot their introduction to orient this work towards phylogenetic studies of “Who Infected Whom” or phylogenetic birth-death processes, where this method seems most useful.

2. In the section “Determining sensitivity and specificity” the discussion of “mutation rate” is confusing. Given that the generation time is the serial interval between infections, the rate at which changes in the genome would accrue AND be observed at the consensus level should be referred to as the pathogen “substitution rate” rather than the “mutation rate”. I realize that may sound pedantic, but this actually caused some confusion for me given that the selected example rate of 1 mutation/genome/generation is actually a reasonable expectation of the biological mutation rate per pathogen replication cycle.

3. I presume that the high substitution rate was selected such that differences in the distributions of expected mutations between linked and unlinked cases (Fig 2B) would appear more distinct. Using genetic distance as the sole basis for distinguishing linked and unlinked cases gets significantly murkier for “natural” substitution rates, as the authors have shown nicely in Fig S4, mentioned on lines 229-230, and discussed in the Discussion. I appreciate those efforts, and I want to stress that I do not feel that this rate selection is disingenuous in any way. However, in the Discussion the authors’ solution to this issue is to incorporate epidemiological data (such as location data, symptom onset date, contact history etc) to improve resolution of linked versus unlinked cases. Again, I don’t deny that multiple data sources would improve these designations, but it is unclear to me then how one would then calculate sensitivity and specificity. Given that this method relies upon knowing those values, this solution actually seems quite challenging to implement and at least mentioning that in the Discussion is important.

4. I find the R_pop quantity to be highly unintuitive. While we generally discuss R_eff as changing over an outbreak given depletion of susceptibles, I’ve never seen a formulation where the average R is calculated across the population with terminal samples presumed to be 0 because their child infections are not sampled. I will say that Figure S2 helped to clarify this concept greatly, and I’m thankful for that addition. However, I still find the in-text explanation (lines 145-157) very confusing. I think the key to making this clearer is to explicitly say that, within the bounded sampling frame, any terminal nodes (leaves) in the tree/transmission network are presumed to have no known child infections, and thus contribute an R value of 0, which is what allows R_pop to drop below one even for diseases where R_eff is easily greater than one.

Reviewer #3: In this work the authors seek to provide guidance to understand how sampling impacts the discovery of transmission events using genomic data. The question is interesting and important but the exploration here is limited to the simplest transmission scenario, with a single introduction, uniform random sampling, a known sensitivity and specificity of the genetic linkage system used (or this can be estimated but again it requires some strong assumptions) and Poisson distributed secondary infections. There is no application to real data, either for a sequenced (or partially sequenced) outbreak with analysis of the study design, or for the exploration of the linkage criteria.

The "single linkage" assumption seems hard to justify and the authors' give a derivation of the main result in S1 Text part D, so it's not clear why this assumption merits so much discussion earlier.

On page 16 of SI Text, k_i is the number of i's true transmission links that are in the sample. So k_i has to add to something less than M, the number of samples. This means that K (sum_i k_i) is not a sum of *independent* Poisson distributed random variables with rate parameter lambda - they are dependent because their sum is constrained. This impacts the expected number of pairs. It would be approximately correct if the sampling fraction is very small, because the sum of k_i would not approach M so the constraint would have minimal impact. But particularly in this paper, something whose bias gets more severe in a way that depends on the sampling fraction is not good. Also the distribution of the number of pairs is important (not just the expectation) .

On the same page I don't get the E(number of true pairs) / Pr(pair is true) - could this be a typo?

- Chi, not X, should be in Table 1

**Have all data underlying the figures and results presented in the manuscript been provided?**

Reviewer #1: Yes

Reviewer #2: None

Reviewer #3: Yes

PLOS authors have the option to publish the peer review history of their article (what does this mean?). If published, this will include your full peer review and any attached files.

Reviewer #1: No

Reviewer #2: No

Reviewer #3: No
---

## [Decision Letter · Decision Letter 1]

20 May 2021

Dear Dr. Lessler,

Thank you very much for submitting your manuscript "Sample Size Calculation for Phylogenetic Case Linkage" for consideration at PLOS Computational Biology. As with all papers reviewed by the journal, your manuscript was reviewed by members of the editorial board and by several independent reviewers. The reviewers appreciated the attention to an important topic. Based on the reviews, we are likely to accept this manuscript for publication, providing that you modify the manuscript according to the review recommendations.

Please address the very minor points raised by the reviewer. Also, note that some of the variables did not render correctly in the pdf of the main text (at least not on my computer). Please check the final submission and ensure that it looks correct. Once these minor points have been addressed, we should be able to accept the manuscript without further review.

Sincerely,

Virginia E. Pitzer, Sc.D.

Deputy Editor-in-Chief

PLOS Computational Biology

Virginia Pitzer

Deputy Editor-in-Chief

PLOS Computational Biology

[LINK]

Please address the very minor points raised by the reviewer. Also, note that some of the variables did not render correctly in the pdf of the main text (at least not on my computer). Please check the final submission and ensure that it looks correct. Once these minor points have been addressed, we should be able to accept the manuscript without further review.

Reviewer's Responses to Questions

**Comments to the Authors:**

Reviewer #1: Many thanks to the authors for considering the points outlined in my previous review. I am satisfied that the authors have adequately addressed all points raised and include only minor typographical feedback below.

Line 136 (marked up version): It may be worth changing mutation to substitution here "rate = 1 mutation/genome/transmission"

Line 281 (marked up version): It might be worth changing the section heading to reflect that it contains multiple examples i.e. "Application to existing datasets"

Lines 386 and 413 (marked up version): The same subheading is used twice for each of the examples, it may be worth making them more specific to the example detailed in each section.

**Have the authors made all data and (if applicable) computational code underlying the findings in their manuscript fully available?**

Reviewer #1: Yes

PLOS authors have the option to publish the peer review history of their article (what does this mean?). If published, this will include your full peer review and any attached files.

Reviewer #1: No

Figure Files:

Data Requirements:

Reproducibility:

References:

---

## [Editor Report · Decision Letter 2]

14 Jun 2021

Dear Dr. Lessler,

We are pleased to inform you that your manuscript 'Sample Size Calculation for Phylogenetic Case Linkage' has been provisionally accepted for publication in PLOS Computational Biology.

Best regards,

Virginia E. Pitzer, Sc.D.

Deputy Editor-in-Chief

PLOS Computational Biology

Virginia Pitzer

Deputy Editor-in-Chief

PLOS Computational Biology

---

## [Editor Report · Acceptance letter]

30 Jun 2021

PCOMPBIOL-D-20-02147R2 

Sample Size Calculation for Phylogenetic Case Linkage

Dear Dr Lessler,

I am pleased to inform you that your manuscript has been formally accepted for publication in PLOS Computational Biology. Your manuscript is now with our production department and you will be notified of the publication date in due course.

With kind regards,

Katalin Szabo
